# Evaluation of Thermal, Hematohistological, and Dermatological Biocompatibility of LED Devices for Neonatal Phototherapy

**DOI:** 10.3390/biomedicines13112826

**Published:** 2025-11-20

**Authors:** Tayomara Ferreira Nascimento, Silvia Cristina Mangini Bocchi, João Cesar Lyra, Rodrigo Fernando Bianchi, Lauro de Assis Duarte Junior, Giselle Silveira Lacerda, Luciana Patrícia Fernandes Abadde, Noeme Sousa Rocha, Susana Eduardo Vieira, Hélio Langoni, Cristiano Neves do Nascimento, Rodrigo Jensen

**Affiliations:** 1Botucatu Medical School, São Paulo State University (UNESP), São Paulo 18618-687, Brazil; silvia.bocchi@unesp.br (S.C.M.B.); joao.lyra@unesp.br (J.C.L.); fernandes.abbade@unesp.br (L.P.F.A.); 2Departamento de Física, Instituto de Ciências Físicas e Biológicas, Federal University of Ouro Preto (UFOP), Minas Gerais 35400-000, Brazil; bianchi@ufop.edu.br (R.F.B.); lauroadjunior@hotmail.com (L.d.A.D.J.); gisellesila@yahoo.com.br (G.S.L.); 3School of Veterinary Medicine and Animal Science, São Paulo State University (UNESP), São Paulo 18618-681, Brazil; noeme.rocha@unesp.br (N.S.R.); susana.vieira@unesp.br (S.E.V.); helio.langoni@unesp.br (H.L.); 4Institute of Bioscience, Botucatu, São Paulo State University (UNESP), São Paulo 18618-689, Brazil; cn.nascimento@unesp.br; 5School of Nursing, University of São Paulo (USP), São Paulo 05403-000, Brazil; rodrigo.jensen@usp.br

**Keywords:** histological analysis, skin response, thermal stability, hematological evaluation, LED therapy, neonatal jaundice

## Abstract

**Background/Objective**: The effectiveness of blue-light phototherapy (PT) is mainly dependent on the total dose of light (time under PT and amount of skin exposed) received by infants. The primary aim of this study was the development of a novel, flexible, and stretchable device to provide continuous PT treatment, avoiding temporary interruptions that are often observed in practice, such as during breastfeeding, for example. This study evaluated the biocompatibility of a novel, low-cost blanket equipped with light-emitting diode (LED) lamps designed to maintain therapeutic efficacy while facilitating uninterrupted skin-to-skin contact. **Methods**: Fourteen New Zealand White rabbits, weighing approximately 2.9 kg and aged 4 months, were randomly assigned to an experimental group (TG, n = 7) or a control group (CG, n = 7). The TG received phototherapy directly on the skin (irradiance: 19.3 [13.0–22.0] µW/cm^−2^/nm^−1^) during two 12 h sessions over consecutive days, while the CG remained under identical conditions with the device turned off. Biochemical, hematological, dermatological, and histological parameters, as well as rectal and skin temperatures, were assessed. **Results**: The results showed no differences in clinical appearance or histological analysis of skin tissue between the groups. Blood analysis indicated a reduction in absolute monocyte counts in the TG compared to the CG (*p =* 0.049), though levels remained within normal ranges. Skin temperature was consistently higher in the TG, except during the initial measurement. Rectal temperatures were similar on the first day but lower in the TG on the second day (mean 40.3 ± 0.21 °C vs. 40.7 ± 0.32 °C; *p =* 0.039). **Conclusions**: Temperature levels remained within physiological limits for both groups throughout the study. The device demonstrated biocompatibility and caused no adverse dermatological, hematological, or biochemical effects.

## 1. Introduction

Phototherapy is an effective treatment for neonatal jaundice or hyperbilirubinemia. This clinical condition affects 60 to 80% of babies and is responsible for numerous hospitalizations [1]. Phototherapy is the most commonly used alternative for neonatal jaundice when there is no risk of kernicterus (severe encephalopathy) and has advantages over other treatments because of its minimal invasiveness [2]. Bilirubin is produced by red blood cell (RBC) metabolism and is normally eliminated from the body through hepatic conjugation with glucuronic acid and excreted in the bile as bilirubin glucuronides. During the normal metabolism process, RBCs from the degradation of the heme group are converted into biliverdin (green color) and subsequently transformed into the bilirubin molecule (yellow color). Bilirubin circulates in the blood bound to serum albumin, is called indirect or unconjugated bilirubin, and is metabolized in the liver [3,4].

In the liver, the enzyme uridine diphosphate glucuronosyltransferase 1A1 (UGT1A1) transforms bilirubin into monoglucuronides and diglucuronide. These water-soluble compounds are excreted in the bile by the transport protein MRP2 [4]. However, in neonates, the hepatic activity of the UGT1A1 enzyme is insufficient, causing bilirubin accumulation and consequently jaundice, also due to the shorter life span of red blood cells at this stage.

Phototherapy consists of exposing the infant to light irradiation, thus preventing bilirubin accumulation and neurotoxicity. Light decomposes the bilirubin molecules and converts them into a water-soluble metabolite that can be eliminated [5].

During phototherapy, light photons act on the bilirubin molecules in nanoseconds, converting them into water-soluble photoproducts of structural isomerization— forming lumirubin, which has an excretion half-life of approximately 1.9 h—and configurational isomerization—bilirubin isomer 4Z, 15E, with an excretion half-life about 13 h. This isomer can be excreted directly, without hepatic metabolism, thus eliminating the metabolite. The existing phototherapeutic systems include sunlight sources, optic fiber blankets, fluorescent lights, halogen spotlights, and light-emitting diode (LED) lamps; the last three systems usually operate from a tripod to photoisomerize bilirubin [6]. Due to environmental pollution, toxicity, and operational lifetime drawbacks, light sources using fluorescent lamps have been replaced by LEDs. When comparing treatments with LED and non-LED devices, LED light therapy has a reduced failure rate, according to data from a systematic review and meta-analysis [6].

Phototherapy conventionally used for neonatal jaundice with remote irradiated lighting affects mothers, as it requires separation from their baby; other drawbacks include continuous crying of the newborn because of the discomfort caused by the lack of clothing and by the eye protection devices required during the treatment, which are stressful for both mother and child [7]. The side effects of phototherapy treatment include hyperthermia and dehydration, mainly caused by lamps that emit heat, especially in tropical countries.

Over the past 70 years, new alternatives have been developed for the management of hyperbilirubinemia, such as blood clearance devices [5], drugs, and devices based on photobiology. In this connection, devices for use close to the human skin, compared to those with light irradiated from a distance, represent advances in terms of humanization of care, increasing parental satisfaction by facilitating bonding, breastfeeding, and the kangaroo method [8].

Technological advances in the manufacture of electronic components and the development of new LED devices and organic light-emitting diodes (OLEDs) for different purposes and applications have enabled the introduction of new forms of light production and delivery devices, such as the use of lamps that do not heat human skin when used at a short distance, enabling more economical, effective, and safer solutions than the use of fluorescent and halogen lamps, since in this case, there is no emission of ultraviolet radiation and there is less heat production [2].

Studies investigating the effects of phototherapy, such as the presence of hydroelectrolytic (dehydration and diarrhea), hematological (change in monocytes), dermatological (skin eruptions), and hyperthermal, have been published [9,10]. Other side effects of phototherapy include chills, eye trauma, increased insensible excretion of water, bronze baby syndrome, eye damage, DNA damage, and nasal obstruction caused by the eye’s protection devices [9,11]. The absorption of water, sodium, and potassium may be impaired in newborns receiving phototherapy, but this effect is transient and resolves after ceasing the treatment. Hematological changes such as leukocyte imbalance, for example, may also occur. Studies have shown an increase in circulating leukocytes after phototherapy [12,13]. Another study corroborates these findings by reporting an increase in the total number of polymorphonuclear cells, such as lymphocytes and monocytes, and arguing that these findings were temporary and without clinical relevance [14]. Dermatological alterations caused by exposure to light as the main triggering factor [12] are also well described.

A review study reports asymptomatic and transient mild or moderate thrombocytopenia (platelet count below 150,000/cm^3^) that can be observed in 79% of newborns after 48 h of phototherapy [15]. Effects of phototherapy on magnesium serum levels (mild reduction in double phototherapy) [16] and on calcium levels (hypocalcemia—serum calcium < 8 mg/dL) have been described in 12.5% of full-term neonates [17].

The use of gonadal protection during phototherapy treatment was also indicated after a single study reported an increased risk of genital squamous cell carcinoma in men with psoriasis treated with psoralens and exposed to UVA/UVB radiation, which has maintained the indication of protection of patients’ genitalia to the present day [18,19]. Considering the above, with a view to observing such clinical variables during treatment with a new LED device proximal to the skin, an experimental rabbit model was set up for the application of the treatment. In order to improve outcomes for patients with clinical problems, multi-method research approaches by nurse scientists and the use of animal models are required. Thus, the animal models used in translational research serve as analogies for clinical problems seen in humans and are therefore useful for investigative advances [20].

This study evaluated a wearable LED device with a flexible structure for neonatal phototherapy in a randomized experimental trial. The primary outcome was the evaluation of 24 h treatment for dermatologic, histologic, hematologic, biochemical, and temperature changes. The secondary outcome measures were the temperature measurements at 30 cm from the experimental area and the temperature per sensor between the device and the skin. Exploratory measures were used to analyze the effects on gonadal tissue.

## 2. Materials and Methods

### 2.1. Wearable Device

The wearable device used in this study measures 15 × 30 cm and has a blue luminous efficiency of 19.3 (13.0–22.0) µw/cm^2^/nm; it was built in accordance with the requirements of the Agencia Nacional de Vigilância Sanitária (ANVISA, National Agency of Sanitary Surveillance) and the Food and Drug Administration (FDA). The wearable device is based on an optimized, transparent, and laminated PVC multilayer film structure and comprises (i) a surface-mounted blue LED emitter layer for standard to intensive phototherapy, (ii) a glycerol-encapsulated bag for rabbit electrical safety, thermal stability, and comfort, and (iii) aluminum foil as a refractive layer (or backlight assembly) for improved lighting conditions. The bags were effective in controlling the temperature of the device, which prevents thermal injury in skin, while the irradiance increases by approximately 20% when the backlight assembly is used. It is a thin, flexible device that can be molded to the body. The tensile stress–strain profile demonstrated that the wearable device had an elongation of approximately two times its original size.

It is covered with human-skin-biocompatible tissue that allows the passage of light. Light emission acts on bilirubin molecules, converting them into water-soluble products (lumirubin and bilirubin isomer 4Z, 15E) that can be excreted directly without the aid of hepatic metabolism. The absorbable light spectrum occurs in the blue spectrum region, close to 460 nm. It is noteworthy that the 460–490 nm spectrum (blue light) is the most efficient spectrum [21]. Blue light was obtained from high-intensity and waterproof flexible LED strip lights (18 units SMD 5050 LEDs, width = 10 mm, voltage = 12 V, beam angle = 120°, operation temperature from 20 to 50 °C, luminous intensity per LED = 800 mcd, wavelength range from 455 to 470 nm, and lifetime = 30,000 h). This device also comprises a refractive (or backlighting) layer for light-intensity enhancement. The multicomponent structure was heat-sealed at the borders and cooled to bond the films and components together to obtain the wearable phototherapy device.

For temperature maintenance and comfort of the rabbits, water bags (or cooling waterbeds to cushion and support the rabbits’ bodies) were obtained by sealing two PVC films by lamination with heat-sealed borders at different temperatures (TS; 85, 100, and 115 °C) after being filled with water and cooled to bond the films together. In the in vitro assessment, the device’s glycerin bag (a component that prevents thermal injuries to the skin) was effective in controlling the temperature resulting from the use of the device [22].

### 2.2. In Vivo Study

A pilot study with three rabbits (not included in the final sample) was conducted prior to the main experiment. The sample consisted of 14 New Zealand White (NZW) rabbits that met the inclusion criteria of being acclimated, weighing more than 1.5 kg, and aged more than 90 days. The pilot study excluded rabbits with pre-existing diseases or skin lesions, and rabbits were housed in individual stainless-steel cages prior to the procedure (for acclimatization). Standard feed and water were provided ad libitum throughout the study period. The cages measured 45 × 60 × 40 cm.

The primary outcome assessed the 24 h treatment for dermatological, histological, hematological, biochemical, and temperature changes. Secondary outcomes comprised temperature measurements taken 30 cm away from the experimental area and the temperature measured by a sensor located between the device and the skin. Exploratory measures were utilized to analyze the effects on gonadal tissue.

### 2.3. Procedure

The rabbits were randomly and individually assigned to one of the numbered cages (Figure 1). Animals were randomized to treatment (TG, n = 7) or control (CG, n = 7) groups using cage number assignment. The rabbits included in the sample were numbered inside the ear from C4 to C17, marked on the inner ear. The ARRIVE 2.0 checklist was used to write this report [23].

### 2.4. Treatment

After light sedation to allow skin shaving, the TG animals underwent phototherapy with an LED device with a mean irradiance of 19.3 (13.0–22.0 µw/cm^2^/nm) in the shaved area located in the right ventrolateral region of the body that was covered by surgical overalls. The treatment was applied for 12 h in a row on Day 1 and for 12 h in a row on Day 2, with a 10 h interval between each session. During the treatment, the animals received water and feed ad libitum. The CG animals received the same procedure adopted for the TG animals, but their device was not connected to the power source, remaining turned off throughout the experiment (sham control).

### 2.5. Temperature Parameters

Skin temperature was monitored using a rectal thermometer and a continuous sensor placed between the shaved skin and the device. The values obtained from both methods were recorded for analysis. Temperature and moisture were properly controlled during the experiment, with the temperature being maintained at 20 °C using wall-split air-conditioning equipment in the room where the rabbits underwent phototherapy.

### 2.6. Serum Collection

Serum collection was performed at three time points as described in the General Principles of Blood Collection in Animals and in the Protocol for Marginal Ear Vein/Artery Blood Sample Collection [24]. The first collection took place before starting the treatment on Day 1, Time Zero (T0), the second in the morning of Day 2, after 12 h of treatment, Time 1 (T1), and the third in the morning after Day 3, Time 2 (T2), after 24 h of treatment. To perform the laboratory tests, a single puncture of the marginal vein was performed, and the material was collected in a 7.5% Ethylene-Diamine-Tetraacetic Acid (K3-EDTA) anticoagulant tube for blood count and a dry tube for biochemical analysis. To perform the blood count of red blood cells, white blood cells, and platelets and to determine hemoglobin, as well as the mean corpuscular volume (MCV), mean corpuscular hemoglobin concentration (MCHC), mean platelet volume (MPV), and red cell distribution width (RDW), the material was placed in an automated veterinary cell counter (impedance). The packed cell volume (PCV) was determined by the Strumia microcapillary method (11,400 rpm for 5 min). The differential leukocyte count, along with the morphological evaluation of red blood cells, white blood cells, and platelets, as well as the estimation of platelets per 1000× field, was performed on blood smears stained with commercial hematological dye (Instant-Prov, Newprov, Pinhais, PR, Brazil), following the recommendations defined by Jain [25]. The platelet count in a hemocytometer was performed according to the methodology described by Brecher et al. [26]. For this purpose, 2 mL of a 1% ammonium oxalate solution was homogenized with 20 µL of blood for five minutes in a vortex. Then, both sides of the hemocytometer were filled, followed by incubation in a moist chamber for 20 min. The platelets contained in the 5 diagonal quadrants of the central grid of the chamber on both sides were counted, and the concentration of platelets/µL was obtained after multiplying by a factor of 2525. For the biochemical analyses, the serum tubes were centrifuged after collection and taken to the laboratory for analysis. The serum biochemistry assessment was performed on automated biochemical equipment (Roche^®^ Cobas Mira Plus, Mannheim, Germany) using kits (Bioclin^®^, Belo Horizonte, MG, Brazil and Labtest^®^, Lagoa Santa, MG, Brazil) according to the manufacturer’s recommendations. Biochemical analyses included alanine aminotransferase (ALT), alkaline phosphatase (AP), gamma-glutamyltransferase (GGT), urea, creatinine, total protein, albumin, and total, indirect, and direct bilirubin [26].

### 2.7. Macroscopic Clinical Evaluation

After shaving the rabbits’ hair, images and a macroscopic evaluation of the skin were obtained before and after each treatment session, at 24, 48, 72, and 96 h. The camera used for the photography was a compact digital Nikon 4300 with automatic mode, macro function activated, and a resolution of 800 × 600 pixels. The classification of skin appearance was conducted in pairs by the investigator and a dermatologist with 24 years of experience in the function, following the adapted Draize Assessment model [27] and photographic record.

### 2.8. Surgical Procedure

Testicular tissue exeresis was performed on Day 4. The animals underwent surgery on a surgical table; they were placed in the supine position for orchiectomy (castration) surgery. The animals were anesthetized with 5% isoflurane, and a vaporizing mask was placed for anesthesia and oxygen supply. After shaving the hair in the inner part of the gonad region, the area was disinfected with 70% alcohol, followed by a 10% povidone iodine solution to prevent infection in the surgical area. A standard approach was used in the right anterior region, exposing the gonad through a 3–4 cm vertical incision for bilateral orchiectomy (castration) surgery. The gonadal tissue was extracted, followed by fixation in 10% neutral buffered formalin for analysis. The incision sites were cleaned with saline and sutured layer-by-layer. Antibiotics and analgesic medications were administered.

### 2.9. Histological Analysis

Rabbits were euthanized on Day 7, with an intravenous overdose of thiopental sodium 150 mg/kg, complying with the American Veterinary Medical Association (AVMA) guidelines [28]. The tissues were then collected (only on Day 7) and dehydrated using a 10% buffered formalin solution for histological evaluation. Samples of skin tissue and gonads were embedded in paraffin and cut into thin sections. Samples were stained with hematoxylin and eosin (H&E) to assess the presence of cellular changes. The slides were mounted, and the images were examined with a conventional 400 × Opticam microscope (Axio Vision Rel 4.8, Carl Zeiss^®^, Carl Zeiss, Germany, high-powered microscope at ×100, ×200, and ×400 magnification) by a senior pathologist with 30 years of experience. The presence or absence of alterations in the epidermis, dermis, muscular morphological structure, intensity and composition of the inflammatory infiltrate, and other findings were described.

### 2.10. Statistical Treatment for Data Analysis

The sample size calculation was based on the incidence of rabbits with a body temperature above 40 °C during 12 h of monitoring. The following assumptions were made: (i) the probability of a rabbit having a body temperature above 40 °C is constant, independent between rabbits, and equal to 0.15; (ii) using a binomial distribution with parameters n and *p* = 0.15, it was estimated that 22 rabbits were required to ensure a 2.8% probability of observing zero rabbits exceeding 40 °C. The experiment’s power was set at 80%, requiring 11 rabbits per group. After the pilot with 3 animals, 22 animals were obtained; however, eight animals died during the habitat transfer and the start of the acclimatization period and were not replaced, leaving 7 rabbits per group. Data collection was stopped for this number of rabbits due to constant repetition of the same responses in the measurements.

Data are expressed as the mean and standard deviation (SD). Confidence intervals of 95% were calculated, and hypothesis tests were performed, setting zero for the mean difference of the values of any hematological and biochemical response variable. Analysis was performed using SPSS version 21.0 (IBM, Armonk, NY, USA). Temperature data were collected longitudinally before and after an intervention, thus allowing observation of the data behavior according to time. Comparisons between groups regarding numerical variables were performed using two-way ANOVA and the Mann–Whitney U test. The results were considered statistically significant at *p* < 0.05. The dataset is available in the Mendeley Data (Digital Commons Data) repository [29].

## 3. Results

### 3.1. Biochemical and Hematological Parameters

Regarding the analysis of the biochemical laboratory parameters, the values collected at times T0, T1, and T2 were within the normal range for rabbits, according to the data presented in Table 1.

When comparing with the Anova model to evaluate at the three time points (T0, T1, and T2), there was a difference in urea (T0 > T1, T2) and creatinine (T1 < T2), as shown in Table 2.

The descriptive analysis, regarding the hematological parameters (Table 3), indicated that the relative values of the parameters between the groups were similar. There was no statistically significant difference in all collections performed at time points T0 and T1; however, at T2, at the end of the treatment, only one statistical difference (*p =* 0.049) in the absolute value of monocytes was observed.

In the T2 evaluation, the ± CG presented higher average absolute values of monocytes (564.1 ± 310.0137) compared to the TG (278.4 ± 152.5766). However, these data do not show statistical significance in the relative values of monocytes.

When comparing the time points before and after treatment for the TG animals, the mean monocyte count decreased from 336.6 cells/mm^3^ before phototherapy to 278.4 cells/mm^3^ after phototherapy.

### 3.2. Dermatological and Histological Parameters of Skin and Gonads

Dermatological evaluations revealed no treatment-associated lesions in either the control (CG, n = 7) or experimental (TG, n = 7) groups. Only one change in the skin was identified, described in Appendix B (Table A2); however, after assessment it was described as just a melanin deposit. The anatomopathological analyses also confirm the findings at the cellular level in both the control (CG, n = 6) and experimental (TG, n = 7) groups, with representative micrographs presented in Figure 2. All animals showed a good general condition, the absence of signs of alterations that could be attributed to the treatment used, and typical hair growth for the species.

As for the evaluation of the gonadal tissue, all samples showed seminiferous tubules and a basal layer made up of delicate connective tissue, followed by germ cells in their different degrees of maturity. Thus, the tissue samples were considered within normality standards.

### 3.3. Thermal Parameters Results

At a distance of 30 cm, the mean temperature ranged from 18.9 ± 0.31 °C to 19.6 ± 1.02 °C. As for the moisture parameters at a 30 cm distance and in the room, measured by a thermo-hygrometer during the 24 h treatment, they varied so that the lowest moisture level recorded 30 cm from the experiment reached 55.4 ± 5.2554 and the mean highest moisture level reached 58.9 ± 6.9864. The lowest moisture level recorded in the experiment room was 55.1 ± 9.8899, and the highest moisture level recorded was 59.7 ± 9.2864.

With the exception of the first assessment (*p =* 0.109), the temperature measured by the sensor showed a statistically significant difference between the two groups, higher in the TG (see Table A1 in Appendix A).

On the first day, the rectal temperature did not differ significantly; however, on Day 2, in the T4 evaluation, the final time point corresponding to the final hours of treatment (Figure 3), it was observed that the temperature was lower in the TG, with a mean of 40.3 (±0.2116) compared to the CG with a mean of 40.7 (±0.3259) (*p* = 0.039).

## 4. Discussion

The blanket employed in this study has been shown to be safe in animal models, thus enabling its translation to clinical studies, as it did not yield significant dermatological, histological, laboratory, biochemical, or hematological alterations.

Previous studies in infants reported that remote LED phototherapy increased eosinophils and basophils, decreased leukocytes and neutrophils, but did not alter monocyte or lymphocyte counts [30]. Our investigation differed in that it used a rabbit animal model as well as a proximal light device; the experiment did not cause hematological alterations in eosinophils and basophils but only in serum monocyte count.

In this connection, these findings suggest that the reduced number of monocytes has minor clinical relevance, since a minimum variation in the hematological parameters is expected and the values are within the reference range for the species. Also, in the groups, the standard deviation is high (high dispersion), and the mean is far apart. In the absolute monocyte count, the difference between the groups was statistically significant. Although the absolute monocyte count showed a statistically significant difference (*p =* 0.049), the relative monocyte values remained similar between groups, suggesting minimal clinical relevance.

Monocytes, once released into circulation, have a half-life of 1 to 2 days due to CCR2/chemokine receptor expression; they respond to this monocyte chemotactic protein and are drawn to sites of tissue injury to participate in inflammatory and phagocytic functions. Experiments indicate that they are predominant in wound infiltrates up to 6 h after muscle injury or tissue trauma. These inflammatory monocytes finally differentiate into classically activated M1 macrophages and, subsequently, into different types of dendritic cells [31]. The histological analyses of the tissues collected, in which cellular inflammatory infiltrates containing macrophages were not found, corroborate our laboratory data. Therefore, these findings suggest that the difference found may not be related to migration to the tissues but to the decrease or maintenance of cell expression (production) or some other factor.

In the literature, one study highlights the analysis of biochemical parameters that present alterations pre- and post-phototherapy in jaundiced neonates [32,33]. The main changes, as a result of the light therapy, are associated with the decline in serum levels of total cholesterol (*p* < 0.05), triglycerides (<0.005), very-low-density lipoprotein (VLDL) (<0.005), uric acid, creatinine, total serum proteins, albumin, and the serum electrolytes sodium, potassium, chloride, and calcium (*p* < 0.001 in each case) [32]. However, in our study, no difference in the biochemical parameter values was found in the TG before and after treatment.

When reviewing the biochemical parameters at T0, T1, and T2, we found a difference in urea (T0 > T1, T2) and creatinine (T1 < T2) when performing repeated measures Anova comparisons (*p* < 0.05).

Regarding dermatological and histological clinical parameters, a new macroscopic finding on the skin was analyzed by histopathological examination; however, it was found to be only a deposit of melanin. Melanocytes make up 1–2% of the epidermal cells, while keratinocytes, which produce keratin, constitute over 95% of the epidermal cells. Melanin absorbs the ultraviolet (UV) photons as well as the free radicals induced by exposure to UV radiation before these free radicals interact with other cellular components. Melanosomes distributed throughout the epidermis provide a highly protective screen that absorbs and scatters harmful UV radiation [31]. It is believed that in this study, the accumulation of skin pigment has arisen as a result of the hair removal and due to exposure to LED light proximal to the skin for 24 h during the treatment and subsequently to the natural lighting of the bioterium facility until euthanasia occurred.

Regarding gonadal pathology, the effect on gonad tissues was an exploratory outcome also investigated in our study. Blinded histopathological analyses performed on rabbit gonadal tissue revealed that the gonadal structures were preserved.

The literature reports only one study that found an increased risk of genital squamous cell carcinoma in humans associated with exposure to PUVA and UVB radiation [18]. As a consequence, gonadal protection during phototherapy has been required. Other experimental studies with newborn rats present results that indicate interference of phototherapy in spermatogenesis, such as a decrease in spermatogonia in the tubules and a decrease in Sertoli cells in the sperm [34,35]. However, these findings were not similar in humans [15].

Regarding the measured temperatures, it was expected that there would be significant differences between the TG with the device turned on and the CG with the device turned off. However, temperature monitoring proximal to the skin revealed that the values remained within normal parameters. Regarding the systemic temperature variation, the groups were similar because, despite the levels being within normal limits in both groups, the difference in the *p*-value of *p =* 0.039 was smaller than the measurement bias of the device, which was ±0.4.

In vitro clinical trials with bilirubin solution and blue light were previously conducted by the authors and described in other studies [36,37]. A recent systematic review and meta-analysis of randomized controlled clinical trials indicated that intermittent blue-light therapy achieves a higher overall efficacy rate and significantly shorter treatment times, with a decrease in the time to clinical resolution of jaundice [38].

It is concluded that the device is biocompatible with the skin and is not the cause of any injury based on an animal model without hyperbilirubinemia. In clinical practice in neonatal patients, the device could perform a function similar to the fur that shields rabbits from cold and heat by acting as a thermal insulator for neonates. It could minimize changes in ambient temperature that may affect infants’ thermoregulation. This is particularly relevant because it is common for newborns, especially premature and low-birth-weight infants, to have difficulty producing enough heat to compensate for heat loss [39].

Reports in the literature indicate the development of recent innovations, with tests still performed in vitro to evaluate wearable OLED devices [40]; however, the LED device tested in our study is a new form of light delivery that has an advantage in that it is low cost.

## 5. Conclusions

The narrowband blue LED wearable device did not cause clinically relevant dermatological, histological, biochemical, or hematological alterations in this non-hyperbilirubinemic rabbit model. These findings suggest that the transfer of understanding and its use in neonatal phototherapy facilities can be performed on an experimental basis. The primary finding was that the blanket had minimal impact on skin temperature. Nevertheless, continuous monitoring of temperature is recommended to ensure treatment safety.

A limitation of this study is that the animals were not submitted to the procedure that causes hyperbilirubinemia. Other laboratory parameters to be assessed in babies that were not controlled in this study are the changes in magnesium, calcium, and vitamin D before and after treatment.

These findings will directly inform hospital-based clinical trials, where the device’s unique design enables therapeutic efficacy while preserving mother–infant bonding during phototherapy. Successful knowledge transfer hinges on (1) optimizing exposure cycles to accommodate breastfeeding intervals, (2) establishing safety thresholds for 72 h continuous use, and (3) validating performance in neonates with hyperbilirubinemia. By overcoming these translational challenges, this technology has the potential to transform neonatal care, replacing conventional isolation-based systems with a biocompatible solution that reduces bilirubin while preserving mother–infant bonding.

## Figures and Tables

**Figure 1 biomedicines-13-02826-f001:**
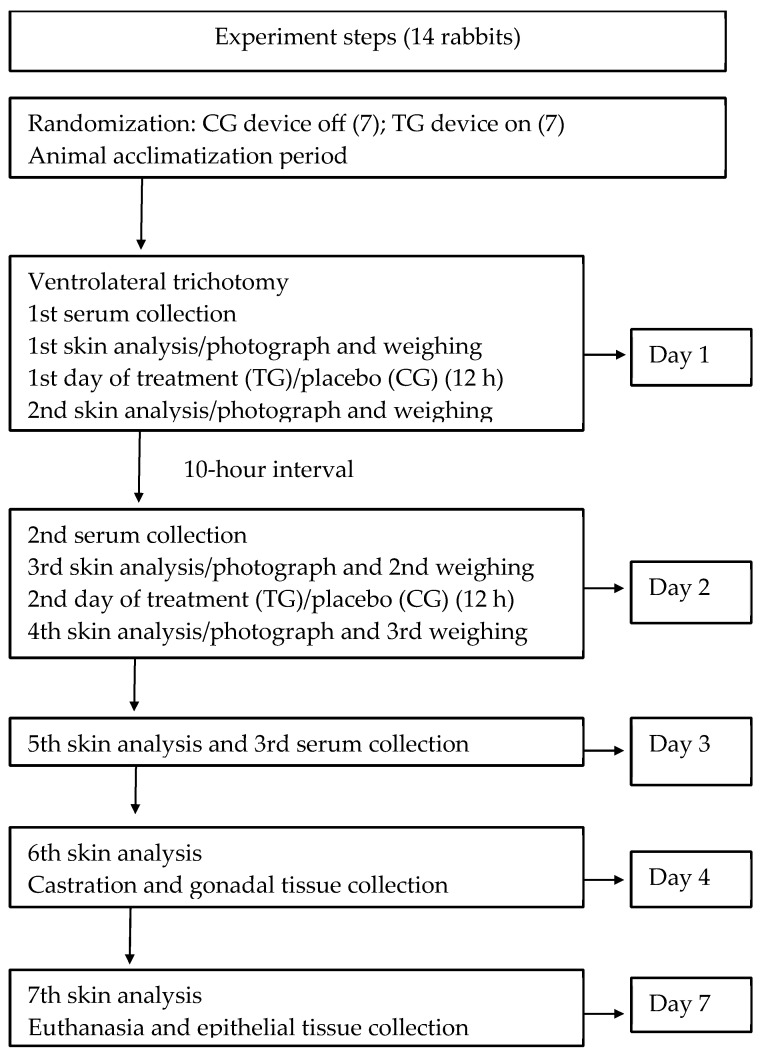
Overview of experimental study protocol. Schematic illustrating the steps of the biocompatibility study of the new LED phototherapy device in the rabbit model. TG: treatment group (LED on, 12 h/day); CG: control group (device off).

**Figure 2 biomedicines-13-02826-f002:**
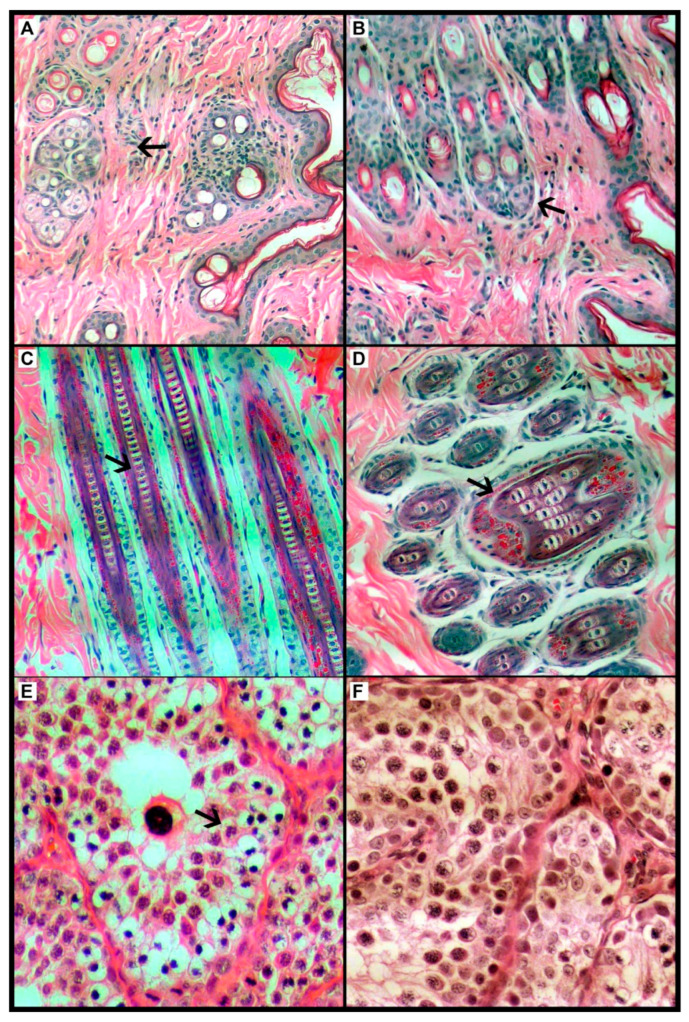
Optical photomicrograph of the Oryctolagus cuniculus rabbit species dermis and epidermis; comparison of control group (**left**) and treatment group (**right**). (**A**) Control group (CG), histological tissue within normal limits, considering the keratin layer, then the epidermis, and finally the cutaneous appendages such as sebaceous and sweat glands and hair follicles intermingled with the supporting connective tissue. (**B**). Treatment group (TG), histological tissue within normal standards, considering the keratin layer, then the epidermal layer and finally the skin appendages such as sebaceous and sweat glands and hair follicles intermingled with the supporting connective tissue. (**C**) CG, the main hair structures can be seen in this sagittal segment, such as the cortex, medulla and cuticle. All are within normal ranges. (**D**). TG, this sagittal segment shows the main hair structures, such as the cortex, medulla, and cuticle. All are within normal ranges. HE stain. Magnification 200×. (**E**) CG, histological aspect of rabbit’s testicle, with focus on one of the seminiferous tubules (arrow). The basal layer consisting of delicate connective tissue, right after the germ cells in their different degrees of maturity. Thus, it was considered within normality standards. (**F**) In the TG, the rabbit testicle is focused on one of the seminiferous tubules (arrow). Basal layer consisting of delicate connective tissue, right after the germ cells in their different degrees of maturity. Thus, it is considered within normality standards. HE stain. Magnification 400×. The arrow indicates the cell structure in (**A**,**B**), the follicle in (**C**,**D**), and the seminiferous tubule in (**E**,**F**).

**Figure 3 biomedicines-13-02826-f003:**
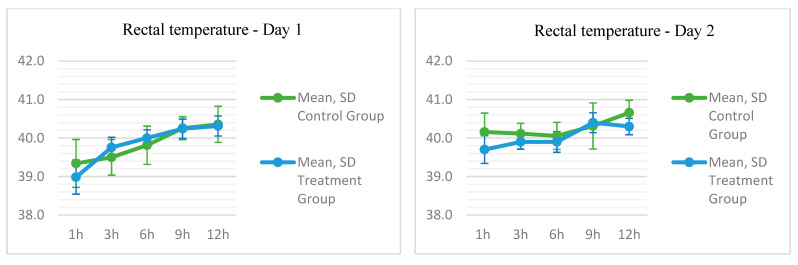
Temperature parameters measured by a rectal thermometer and sensor in New Zealand rabbits submitted to phototherapy treatment using an LED device.

**Table 1 biomedicines-13-02826-t001:** Biochemical parameters of New Zealand rabbits submitted to phototherapy treatment using an LED device.

Measure	Before Starting the Experiment (T0)		
	Control (n 7)	Treatment (n 7)	*p*-Value
	M SD	M SD	
Urea (mg/dL^−1^)	38.1 ± 7.9042	31.1 ± 4.0178	0.059
Creatinine (mg/dL^−1^)	0.8 ± 0.1604	0.7 ± 0.0770	0.249
ALT—alanine aminotransferase (UI-L)	51.6 ± 10.7371	54.9 ± 15.3126	0.650
ALP—alkaline phosphatase (UI-L)	165.9 ± 66.5919	177.6 ± 30.7401	0.680
Gamma glutamyl transferase (UI-L)	3.7 ± 2.6475	4.5 ± 1.5703	0.475
Serum total protein (UI-L)	5.8 ± 0.1397	5.7 ± 0.4923	0.566
Albumin (gdL^−1^)	4.0 ± 0.2440	4.1 ± 0.1864	0.550
Globulin (gdL-1)	1.8 ± 0.3309	1.6 ± 0.3891	0.259
Total bilirubin (mg/dL^−1^)	0.1 ± 0.0549	0.1 ± 0.0787	0.276
Direct bilirubin (mg/dL^−1^)	0.0 ± 0.0488	0.0 ± 0.0378	0.552
Indirect bilirubin (mg/dL^−1^)	0.1 ± 0.0756	0.1 ± 0.0951	0.237
Measure	After treatment (T2)		
Urea (mg/dL^−1^)	32.9 ± 8.3152	28.4 ± 5.1916	0.255
Creatinine (mg/dL^−1^)	0.8 ± 0.1941	0.8 ± 0.1243	0.577
ALT—alanine aminotransferase (UI-L)	58.0 ± 14.8773	59.4 ± 13.6364	0.855
ALP—alkaline phosphatase (UI-L)	158.3 ± 44.4512	195.7 ± 46.8640	0.151
Gamma glutamyl transferase (UI-L)	3.9 ± 2.5844	4.5 ± 1.6359	0.622
Serum total protein (UI-L)	6.0 ± 0.3101	5.9 ± 0.5460	0.598
Albumin (gdL^−1^)	4.2 ± 0.2289	4.2 ± 0.2160	0.814
Globulin (gdL^−1^)	1.6 ± 0.7616	1.7 ± 0.4276	0.865
Total bilirubin (mg/dL^−1^)	0.1 ± 0.0378	0.1 ± 0.0690	0.356
Direct bilirubin (mg/dL^−1^)	0.0 ± 0.0378	0.0 ± 0.0690	1.000
Indirect bilirubin (mg/dL^−1^)	0.1 ± 0.0577	0.1 ± 0.0577	1.000

Note: M = mean; SD = standard deviation.

**Table 2 biomedicines-13-02826-t002:** Multiple comparisons of biochemical parameters of New Zealand rabbits submitted to phototherapy treatment using an LED device.

Variable	pMauchly	pANOVA	Multiple Comparisons
Urea (mg/dL^−1^)	0.073	<0.001	M0 > M1, M2
Creatinine (mg/dL^−1^)	0.25	0.031	M1 < M2
ALT—alanine aminotransferase (UI-L)	0.183	0.066	M0 = M1 = M2
ALP—alkaline phosphatase (UI-L)	0.454	0.806	M0 = M1 = M2
Gamma glutamyl transferase (UI-L)	0.087	0.291	M0 = M1 = M2
Serum total protein (UI-L)	<0.001	0.352	M0 = M1 = M2
Albumin (gdL^−1^)	<0.001	0.330	M0 = M1 = M2
Globulin (gdL^−1^)	<0.001	0.420	M0 = M1 = M2
Total bilirubin (mg/dL^−1^)	0.634	0.546	M0 = M1 = M2
Direct bilirubin (mg/dL^−1^)	0.022	0.706	M0 = M1 = M2
Indirect bilirubin (mg/dL^−1^)	0.158	0.753	M0 = M1 = M2

Note: pANOVA = *p*-value of the ANOVA model for repeated measures; pMauchly = Mauchly’s test of sphericity (*p*-value).

**Table 3 biomedicines-13-02826-t003:** Hematological parameters of New Zealand rabbits submitted to phototherapy treatment using an LED device.

Measure	Before (T0)		
	Control (n 7 ^a^)	Treatment (n 7)	*p*-Value
	M SD	M SD	
Red blood cells (µL)	5.9 ± 0.6770	5.9 ± 0.6625	0.991
Hemoglobin (g/dL)	11.4 ± 0.9661	11.6 ± 1.1838	0.753
Hematocrit (%)	39.3 ± 2.6277	40.0 ± 3.0000	0.644
MCV (fL)	66.8 ± 5.3232	67.9 ± 5.0340	0.692
MCHC (%)	29.1 ± 2.8644	29.0 ± 2.4478	0.938
PT (Plasma) (g/dL)	5.9 ± 0.2545	5.9 ± 0.4860	0.788
RDW (%) ^a^	14.8 ± 1.6669	14.9 ± 1.5394	0.930
Platelets (µL) (10^5^)	3.17 ± 0.64	3.73 ± 0.10	0.289
Leukocytes (µL)	7814.3 ± 5148.2776	6557.1 ± 1739.5949	0.552
Relative neutrophils (%)	41.3 ± 17.4520	45.0 ± 12.8452	0.658
Absolute neutrophils (µL)	3804.7 ± 4231.2037	3011.4 ± 1505.5346	0.649
Relat lymphocytes (%)	51.3 ± 17.2889	47.4 ± 15.1091	0.665
Abs lymphocytes (µL)	3515.0 ± 1331.9323	3084.7 ± 1187.8752	0.536
Relat eosinophils (%)	1.7 ± 0.7559	2.3 ± 1.6036	0.410
Abs eosinophils (µL)	115.7 ± 53.3439	145.0 ± 125.1919	0.580
Relat basophils (%)	0.1 ± 0.3780	0.0 ± 0.0000	0.337
Abs basophils (µL)	4.6 ± 12.0949	0.0 ± 0.0000	0.337
Relat monocytes (%)	5.4 ± 4.2762	5.6 ± 3.9940	0.950
Abs monocytes (µL)	363.7 ± 358.9562	336.6 ± 167.4344	0.859
Measure	After (T2)		
Red blood cells (µL)	6.2 ± 0.5029	6.1 ± 0.2829	0.858
Hemoglobin (g/dL)	12.1 ± 0.6630	12.1 ± 0.8789	0.973
Hematocrit (%)	40.9 ± 2.1931	40.7 ± 1.7043	0.894
MCV (fL)	66.5 ± 3.9352	66.5 ± 2.9648	0.975
MCHC (%)	29.8 ± 2.0244	29.8 ± 1.5261	0.999
PT (Plasma) (g/dL)	6.2 ± 0.1799	6.0 ± 0.5589	0.532
RDW (%) ^b^	14.4 ± 1.1545	15.0 ± 1.4318	0.438
Platelets (µL) (10^5^)	327,285.7 ± 98,915.8037	359,878.6 ± 63,324.7359	0.477
Leukocytes (µL)	7371.4 ± 3233.0142	6028.6 ± 832.0943	0.308
Relative segmented (%)	42.1 ± 13.2844	49.4 ± 11.2229	0.289
Absolute segmented (µL)	3368.0 ± 2695.3135	3015.0 ± 918.9525	0.749
Relative lymphocytes (%)	48.9 ± 13.2467	44.0 ± 11.3578	0.476
Absolute lymphocytes (µL)	3339.1 ± 701.1746	2641.0 ± 732.8708	0.094
Relative eosinophils (%)	1.1 ± 1.4639	1.7 ± 1.7995	0.527
Absolute eosinophils (µL)	100.1 ± 158.5564	114.4 ± 94.9664	0.841
Relative basophils (%)	0.0 ± 0.0000	0.0 ± 0.0000	
Absolute basophils (µL)	0.0 ± 0.0000	0.0 ± 0.0000	
Relative monocytes (%)	7.9 ± 4.4132	4.9 ± 3.1320	0.168
Absolute monocytes (µL)	564.1 ± 310.0137 ^b^	278.4 ± 152.5766 ^b^	0.049 ^b^

Note: ^a^ (n6) in both groups, treatment and control C7 and C8, difficult collection and inappropriate sample for analysis. ^b^ = means with letter “b” differ statistically from each other.

## Data Availability

The dataset is available in the Mendeley Data Repository [29].

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
