# Peer review of "Evaluation of Thermal, Hematohistological, and Dermatological Biocompatibility of LED Devices for Neonatal Phototherapy"

_biomedicines, 2025, doi:10.3390/biomedicines13112826_

Round 1

Reviewer 1 Report

Comments and Suggestions for Authors

In this study, the authors report wearable sensors

and wearable LED devices for bio-monitoring

applications. The manuscript can be recommended 

for publication following some minor revisions.

  1. The sensor may be a good application. The sensor parameters should be described in detail
  2.  What are the sensing mechanisms
  3. The sensor fabrication be explained well, e.g., materials used, methods, schematic illustrations etc.
  4. The sensors should be strain sensing to expand the scope of this research 

Author Response

Reviewer 1

Comment 1: The sensor may be a good application. The sensor parameters should be described in detail. Response: The device consists of a multilayered film structure in which a multicomponent arrangement (refractive layer, LED illumination unit, and safety bags) is laminated between the bottom and top layers of transparent, biocompatible, and mechanically robust poly(vinyl chloride) (PVC). Water bags were sealed using heat-laminated borders at 85, 100, and 115 °C, filled with water, and cooled to ensure adhesion and durability.

Comment 2: What are the sensing mechanisms? Response: Blue light was obtained from high-intensity waterproof LED strips (18 SMD 5050 LEDs, width = 10 mm, voltage = 12 V, beam angle = 120°, operating temperature 20–50 °C, luminous intensity = 800 mcd, wavelength 455–470 nm, lifetime 30,000 h). The device includes a refractive (backlighting) layer to enhance light intensity. The multilayer structure was heat-sealed and cooled to bond all components into a wearable phototherapy device.

Comment 3: Sensor fabrication should be explained (materials, methods, schematic illustrations, etc.). Response: The device comprises:
(i) Surface-mounted blue LED emitter layer for standard to intensive phototherapy.
(ii) Glycerol-encapsulated bag ensuring electrical safety, thermal stability, and comfort.
(iii) Aluminum foil backlighting assembly to enhance irradiance (~20%).

The device is thin, flexible, and body-conformable. Tensile stress–strain analysis confirmed elongation of ~2× its original size.

Reviewer 2 Report

Comments and Suggestions for Authors

This study evaluated the biocompatibility of a novel, low-cost, wearable LED blanket for potential use in neonatal phototherapy using a rabbit model. The device demonstrated thermal stability and caused no clinically relevant dermatological, hematological, or biochemical changes, supporting its safe application in future clinical settings. I commend the authors for this important contribution and offer the following suggestions to improve the manuscript:

  1. Please ensure that “LED” is capitalized consistently in the title and throughout the manuscript.
  2. In the abstract, the Background/Objectives section should be expanded to include one or two sentences outlining the clinical relevance and current limitations of existing phototherapy methods to provide better context for the study.
  3. The term “Light Emitting Diode” does not need to be capitalized unless it begins a sentence. Please revise accordingly.
  4. P-values should be formatted consistently using leading zeros (e.g., p = 0.049 rather than p = .049).
  5. The authors may consider emphasizing parameters that showed statistically significant changes (e.g., monocyte count, rectal temperature) to facilitate easier identification by readers.
  6. The manuscript should clarify the time point at which histological analyses were performed. If multiple time points were evaluated, please include and discuss the findings.
  7. Please indicate the number of samples used for histological examination and whether the images shown are representative. Additionally, scale bars should be added to all histological images.
  8. In Figure 3, the phrase “Botucatu, SP, Brazil, 2021” in the caption is unclear. Please explain its relevance or remove it if unnecessary. Also, do the authors have any data from time points beyond day 2?
  9. The potential impact of the device and procedures on animal well-being—particularly with repeated use or long-term exposure—should be discussed to better frame its translational relevance.

Author Response

Reviewer 2

Comment 1: LED capitalization. Response: Corrected throughout the manuscript.

Comment 2: Expand clinical relevance in the Abstract. Response: The effectiveness of blue-light phototherapy (PT) depends on exposure duration and skin area illuminated. Conventional PT often requires interruptions (e.g., breastfeeding), reducing treatment efficiency. This study aimed to develop a flexible, continuous PT device to minimize interruptions and preserve maternal–infant bonding.

Comment 3: Light Emitting Diode capitalization. Response: Corrected (capitalized only at sentence start).

Comment 4: P-value formatting. Response: Corrected to include leading zeros (e.g., p = 0.049).

Comment 5: Highlight statistically significant parameters. Response: Significant parameters such as monocyte count and rectal temperature are now emphasized in the Results section.

Comment 6: Histological analysis time point. Response: Analyses were performed on Day 7 at euthanasia (Section 2.6).

Comment 7: Sample number and scale bars. Response: Dermatological evaluation included CG (n = 7) and TG (n = 7). Scale bars were added to all histological figures; images are representative.

Comment 8: Figure 3 caption (“Botucatu, SP, Brazil, 2021”). Response: Removed. No additional time points beyond Day 2 were collected.

Comment 9: Translational relevance and animal welfare. Response: These findings support clinical translation, allowing therapeutic efficacy while preserving maternal–infant bonding. Future clinical studies should consider:
1. Optimizing exposure cycles for breastfeeding intervals.
2. Establishing safety thresholds for up to 72 h continuous use.
3. Validating performance in neonates with hyperbilirubinemia.

Reviewer 3 Report

Comments and Suggestions for Authors

In this manuscript, the researchers investigate the biocompatibility of a blue LED wearable device from biochemical, hematological, dermatological, and histological perspectives, and validate the safety of this device at the animal (rabbit) level, thereby providing valuable reference for its application in the treatment of neonatal jaundice or hyperbilirubinemia; however, before this manuscript can be accepted, please address the following issues:

  1. For Section 3.1 “Biochemical and hematological parameters,” why are the T1 data not provided in Table 1 and Table 3, yet T1 comparisons appear in Table 2? Please either supply the missing T1 data or explain the reason for omitting T1.

  2. Although the authors state in the Conclusion that “the animals were not submitted to the procedure that causes hyperbilirubinemia,” I am still concerned that normal skin tissue and jaundiced (hyperbilirubinemic) lesioned skin tissue absorb 460–490 nm light with different efficiencies, leading to differing photothermal effects in the tissue. If only healthy rabbits are used to test the device’s biocompatibility, might this weaken the clinical significance of the paper? If possible, please consider supplementing with in vitro cellular safety tests (e.g., in a bilirubin solution), or else reorganize the manuscript to focus just on the device’s excellent biosafety on normal skin tissue.

  3. Other minor issues:

    Line 134: “19.3 (13.0–22.0) µw/cm²/nm;” here “µw” should be written as “µW”;

    Line 351: the notation “(p = 1.09)” should be checked;

    Line 339: please verify and correct the subsection numbering for “Temperature parameters.”

Thank you!

Author Response

Comment 1: Missing T1 data in Tables 1 and 3. Response: T1 data were initially omitted to emphasize before-and-after treatment differences. T1 values are now included in the revised tables for clarity. The dataset is available in the Mendeley Data Repository [29]. https://doi.org/10.17632/s56gr964h6.1.

Comment 2: Concern about testing only in healthy animals. Response: The study establishes biocompatibility in healthy skin. We acknowledge that bilirubin absorption may alter photothermal effects in hyperbilirubinemic conditions. Previous in vitro studies with bilirubin solutions [36,37] and a systematic review/meta-analysis of 63 RCTs [38] corroborate the safety and efficacy of blue-light phototherapy. Further in vivo studies in hyperbilirubinemic models are warranted.

Comment 3: Minor corrections. - Line 134: “µw/cm²/nm” → µW/cm²/nm (3 locations)
- Line 351: “p = 1.09” → p = 0.109
- Line 339: corrected subsection numbering; 

Round 2

Reviewer 2 Report

Comments and Suggestions for Authors

Scale bar is still missing for the H&E images. 

Author Response

Comment 1: The microscope on which the images were obtained did not have a scale bar, and it was not possible to obtain new photos with the same image quality. Therefore, we chose not to replace them.

Reviewer 3 Report

Comments and Suggestions for Authors

The authors have thoroughly addressed the issues raised in the prior review and have basically resolved the outstanding concerns. After correction of a small number of minor matters (e.g. typographical errors and minor formatting/language edits), the manuscript is suitable for publication in this journal.

Author Response

Comments 1: I thank the reviewer for their review work.